# Construction of Cisplatin-18-Crown-6 Complexes Through Supramolecular Chemistry to Improve Solubility, Stability, and Antitumor Activity

**DOI:** 10.3390/ijms252413411

**Published:** 2024-12-14

**Authors:** Yue Gao, Yeqi Huang, Chuanyu Ren, Si Xiong, Xia Guo, Ziyu Zhao, Ling Guo, Zhengwei Huang

**Affiliations:** 1Department of Pharmacy, College of Pharmacy, Jinan University, Guangzhou 511436, China; gaoyue122612@stu.jnu.edu.cn (Y.G.); yeqihuang@stu.jnu.edu.cn (Y.H.); r1094243897@stu.jnu.edu.cn (C.R.); 2State Key Laboratory of Bioactive Molecules and Druggability Assessment, Guangdong Basic Research Center of Excellence for Natural Bioactive Molecules and Discovery of Innovative Drugs, College of Pharmacy, Jinan University, Guangzhou 511436, China; xiongsi123@jnu.edu.cn (S.X.); xiaguo119@jnu.edu.cn (X.G.); 3Key Laboratory of Tropical Biological Resources of Ministry of Education, School of Pharmaceutical Sciences, Hainan University, Haikou 570228, China; guoling@hainanu.edu.cn

**Keywords:** cisplatin, 18-crown-6, supramolecular chemistry, complexes, solubility, stability, cytotoxicity

## Abstract

Cisplatin (DDP), a platinum-chelated compound renowned for its antitumor activity, is often utilized in cancer therapy. However, its real-world clinical efficacy is compromised by poor solubility and low stability, which impedes wider clinical application. Our study aimed to address these limitations of DDP through host–guest supramolecular chemistry approaches. We explored the potential of 18-crown-6 as the host molecule to solubilize and stabilize DDP, the guest molecule. Utilizing techniques such as UV–visible spectroscopy, Fourier-transform infrared spectroscopy, Raman spectroscopy, and molecular docking, we conducted a comprehensive analysis on the physical state and inclusion mode of the DDP@18-crown-6 complex. Phase solubility studies and Job’s plot confirmed that the DDP@18-crown-6 complex significantly enhanced the aqueous solubility of DDP, with an optimal 1:1 binding ratio. Stability analyses revealed that this complex markedly improved the stability of DDP in pure water. Meanwhile, the stabilization effects of DDP@18-crown-6 were remarkably elevated when combined with 0.9% sodium chloride. In vitro antitumor assays in A549 cell lines demonstrated that the DDP@18-crown-6 complex outperformed raw DDP in cytotoxicity, showing a significantly lower IC_50_ value. This research offered a promising strategy for DDP solubilization and stabilization, facilitating its anticancer therapeutic efficacy.

## 1. Introduction

Lung cancer, a malignancy originating in the lungs, is predominantly linked to smoking, environmental pollutants, and genetic predispositions [1,2]. As one of the most prevalent cancers globally and a leading cause of cancer-related mortality, it poses a significant threat to human health and safety [3]. As a platinum (Pt)-chelated drug, DDP exerts its antitumor effect by forming DNA cross-links, thereby hindering the replication and repair of cancer cells and disrupting tumor growth [4].

However, two key challenges currently limit the wider clinical application of DDP: (1) DDP exhibits poor aqueous solubility, with an intrinsic solubility of approximately 1 mg/mL in pure water, complicating formulation development, especially for high-dosage formulations [5,6]. (2) The storage stability and photostability of DDP are notably compromised. In aqueous environment, DDP is highly unstable, with chloride ions (Cl^−^) readily replaced by water molecules, leading to loss of activity. To enhance its stability and bioactivity, DDP is typically dissolved in 0.9% sodium chloride (NaCl) solution. Nevertheless, this method only preserves activity temporarily, without long-term effects [7,8]. DDP also undergoes photohydrolysis and photodegradation under light irradiation, resulting in Pt precipitation. Thus, strict light avoidance is essential during storage and use [9]. These solubility and instability issues significantly impede the storage and utilization of DDP [5]. Consequently, there is an urgent need for strategies to address these challenges.

Supramolecular chemistry approaches are promising to fulfill such a goal. This is a field pioneered by French chemist Dr. Jean-Marie Lehn, which explores the interactions between two or more chemical entities through non-covalent forces—such as hydrogen bonds, electrostatic interactions, ion–π interactions, hydrophobic effects, π–π stacking, and coordination bonds—to form complex systems with defined structures and functions [10]. Host–guest chemistry is the major domain of supramolecular chemistry. Host–guest chemistry focuses on supramolecular entities based on host molecules like crown ethers [11,12], cyclodextrins [13], calixarenes [14], pillararenes [15], and cucurbiturils [16], which form complexes with guest molecules via non-covalent bonds (predominantly hydrogen bonding). Notably, these complexes address issues related to drug solubility and stability, highlighting their research significance in pharmaceutical development.

Among various host molecules, 18-crown-6 has garnered substantial attention due to its unique macrocyclic ether structure. Its molecular structure is depicted in Figure 1B, which features a macrocyclic structure composed of six ethylene oxide units (-O-CH_2_-CH_2_-) [17]. This molecule forms a slightly curved ring where the oxygen and carbon atoms do not lie in a single plane, resembling a “crown” in shape (Figure 1C) [18]. Its open-ring structure provides a relatively uniform cavity of 4.3 Å, which accommodates molecules or ions of appropriate size and shape [15,19,20]. This spatial configuration suggests promising applications for 18-crown-6 as drug delivery systems. Recently, Chai and colleagues [21] explored the feasibility of synthesizing an 18-crown-6-based polymer incorporating curcumin. Their findings demonstrated that this synthesized polymer effectively addressed the poor water solubility and low bioavailability of curcumin while enhancing its anticancer activity.

Noticeably, although 18-crown-6 is overall neutral in charge, the presence of six oxygen atoms creates a polar region with localized negative charge density [22]. The unique charge distribution allows 18-crown-6 to serve as a potential host for guests with positive charges, like metallic compounds [23]. This is regarded as the characteristic advantage of 18-crown-6 beyond other host molecules. Jing et al. used density functional theory and molecular dynamics simulation to evaluate the selection and loading efficiency of 18-crown-6 on metallic compounds (Li^+^, Na^+^, K^+^, Rb^+^, Cs^+^). The experiments indicated that 18-crown-6 could cooperate with these ions through non-covalent interactions. Simultaneously, in comparison with other ions, K^+^ fitted 18-crown-6 more due to its ionic diameter, thereby having stronger binding and selection with the subject [24]. Consequently, 18-crown-6 holds promise as a suitable solubilizer and stabilizer to overcome the current limitations of the Pt-containing DDP, potentially boosting its anticancer efficacy.

We synthesized the DDP@18-crown-6 complex to validate these improvements, optimizing its preparation conditions. The complex was characterized through UV–visible spectroscopy, Fourier-transform infrared (FT-IR) spectroscopy, Raman spectroscopy, and molecular docking, confirming the successful formation. Phase solubility studies and Job’s plot were employed to assess the impact of 18-crown-6 on DDP solubility and the optimal binding ratio of the two molecules. Additionally, we evaluated the complex’s effects on DDP stability (such as time-evolution stability and photostability) and estimated the impact of different concentrations of 18-crown-6 on its stability. Compared to the raw DDP, the anti-human-lung-cancer cell activity of the complex was assayed by cell counting kit-8 (CCK8) assay. The efficacy of the present study in the enhancement of water solubility and stability of DDP provides a rational basis for solubilization and use of drugs with low water solubility and poor stability. Moreover, the employment of supramolecular chemistry to enhance drug activity promises to be a powerful solution to the problems of drug solubility and stability in cancer therapy and even in the treatment of other diseases.

## 2. Results and Discussion

### 2.1. Preparation of DDP@18-Crown-6 Complex

DDP was configured as an aqueous solution at various concentrations. As illustrated in Appendix A, the concentration of the DDP solution exhibited a direct correlation with absorbance at 301 nm. The resulting linear regression analysis yielded the equation *Y* = 0.3459*X* + 0.0827 (*R*^2^ > 0.99). Upon the addition of 18-crown-6 to the DDP solution, a potential increase in solubility was observed, likely due to the formation of a complex between DDP and 18-crown-6. Absorbance measurements of the resulting complex system were then performed, and the standard curve equation was applied to assess the impact of the DDP@18-crown-6 complex on the solubility enhancement of DDP.

The duration of complex preparation markedly influences the binding affinity between DDP and 18-crown-6. Optimal preparation time was thus evaluated [25]. As depicted in Figure 2A and Appendix A, the DDP@18-crown-6 complex exhibited peak UV absorbance at 301 nm after 48 h of mixing and the highest concentration of DDP in the solution (*p* < 0.0001). Prolonged preparation times resulted in decreased absorbance, likely due to drug degradation from excessive mechanical stirring [26]. Therefore, the ideal preparation time for the DDP@18-crown-6 complex was set as 48 h. Additionally, the stirring rate significantly affects the binding efficiency between DDP and 18-crown-6, necessitating careful optimization of this parameter during complex formation [25]. As shown in Figure 2B and Appendix A, the complex solution achieved peak UV absorbance and maximum DDP concentration at a stirring rate of 1200 rpm. To ensure optimal binding between DDP and 18-crown-6 and maximize DDP solubility, 1200 rpm was chosen as the optimal stirring rate. Thus, the ideal conditions for preparing the DDP@18-crown-6 complex were 48 h of magnetic stirring at 1200 rpm. An intriguing test of the solubilization effects was performed. Excessive DDP could hardly be dissolved in ultrapure water, and yellowish precipitate was found, but a clear solution was witnessed which consisted of the same amount of DDP and an equimolar solution of 18-crown-6 (Figure 2C), demonstrating enhanced solubility of DDP in the 18-crown-6 solution. These results preliminarily indicated that the addition of 18-crown-6 significantly improves the solubility of DDP in aqueous systems.

### 2.2. UV–Visible Spectral Analysis

UV–visible spectroscopy is a powerful tool for the qualitative and quantitative analysis of the chemical structures and interactions, offering high sensitivity and accuracy [27]. By analyzing the λ values of absorption peaks, one can effectively identify compound structures, while variations in peak intensity or absorbance enable quantitative assessment of potential interactions [28]. To ascertain the formation of the DDP@18-crown-6 complex and its solubilizing effect, we investigated the UV–visible spectrogram of DDP, 18-crown-6, and the DDP@18-crown-6 complex (Figure 3). The absence of UV absorbance peaks for 18-crown-6 in the investigated ranges indicated that it did not interfere with DDP’s absorbance; the spectroscopic profile of DDP was credible. DDP exhibited a characteristic peak at 301 nm. The DDP@18-crown-6 complex showed a pronounced enhancement in absorbance at 301 nm compared to pure DDP, confirming that 18-crown-6 facilitated greater solubilization of DDP and revealed a higher level of UV absorbance. These UV data suggested that DDP and 18-crown-6 likely form a complex, significantly improving the solubility of DDP in aqueous solutions.

### 2.3. FT-IR Analysis

FT-IR spectroscopy is another widely employed analytical technique, offering high resolution and sensitivity to elucidate chemical structures and reaction mechanisms [29]. In FT-IR spectra, specific peak shifts correspond to distinct functional groups and chemical bonds [30]. This method allows for the investigation of interactions between DDP and 18-crown-6. The FT-IR spectra of the three compounds (Figure 4A) revealed a peak for saturated C-H stretching at 2896 cm^−1^ of the DDP@18-crown-6 complex. Additionally, C-O stretching vibrations at 1288 cm^−1^ and 1245 cm^−1^ confirmed the presence of intact 18-crown-6 molecules within the complex [31]. Furthermore, the FT-IR spectrum of DDP revealed three characteristic peaks at 3290 cm^−1^, 1285 cm^−1^, and 796 cm^−1^ [32]. Nevertheless, in the spectrum of the DDP@18-crown-6 complex, the peaks at 1285 cm^−1^ and 796 cm^−1^ were obscured by the C-O stretching and C-H bending vibrations of 18-crown-6. Notably, the N-H stretch peak shifted from 3290 cm^−1^ to 3274 cm^−1^, indicating a redshift (Δ = 16 cm^−1^). Similarly, the C-O stretching peaks shifted from 1296 cm^−1^ and 1259 cm^−1^ in 18-crown-6 to 1283 cm^−1^ (Δ = 16 cm^−1^) and 1245 cm^−1^ (Δ = 14 cm^−1^) in the complex [33]. These shifts suggested that hydrogen bonding occurred between the -NH_3_ groups of DDP and the oxygen atoms of 18-crown-6.

### 2.4. Raman Spectroscopic Analysis

Raman spectroscopy, a technique based on light scattering, is pivotal for the characterization and study of molecular structures, often acting as a supplement for FT-IR. The distinctive Raman peaks provide critical insights into drug molecule identification, with peak shifts revealing specific binding interactions [34]. This method was employed to investigate the interaction between DDP and 18-crown-6. Raman spectra (Figure 4B) showed an asymmetric ν(Pt-Cl) stretch at 323 cm^−1^ in DDP, with symmetric NH_3_ features at 1316 cm^−1^ and 1295 cm^−1^ [35]. The bands at 522 cm^−1^ and 507 cm^−1^ corresponded to symmetric and asymmetric ν(Pt-NH_3_) stretches, respectively [35]. Notably, in the complex, the ν(Pt-NH_3_) stretches redshifted to 545 cm^−1^ and 531 cm^−1^ (Δ = 16 cm^−1^), likely due to hydrogen bonding interactions between 18-crown-6 and the Pt-NH_3_ groups in DDP. The Raman spectra revealed a notable shift in the symmetric NH_3_ peaks to 1271 cm^−1^ (Δ = 45 cm^−1^) and 1241 cm^−1^ (Δ = 54 cm^−1^), consistent with previous observations. The presence of the ν(Pt-Cl) stretching vibration in the complex confirmed the structural integrity of DDP, with no hydrolysis occurring [36]. Furthermore, a marked blue shift in the ν(C-O) vibration (1132 cm^−1^ vs. 1145 cm^−1^, Δ = 13 cm^−1^) relative to the 18-crown-6 spectrum validated that the hydrogen bonding interaction occurred between the oxygen atoms of 18-crown-6 and DDP [37], aligning with FT-IR spectral findings.

### 2.5. Molecular Docking

Molecular docking is a crucial computational method used to explore the interaction sites and mechanisms between molecules. This technique optimizes the conformations, dihedral angles, and positions of rotatable bonds to identify the most favorable binding configurations [38,39].

The optimal binding conformation of DDP and 18-crown-6 is illustrated in Figure 5. Specifically, the interaction involved hydrogen bonding between the -NH_3_ ligands of DDP and the oxygen atoms of 18-crown-6, forming a hydrogen bonding network (indicated by blue dashed lines) and resulting in the DDP@18-crown-6 complex. This configuration aligned with the above experimental observations. The hydrogen bonding bond lengths between the two molecules (ranging from 2.1 to 2.8 Å) are labeled on the blue dashed line in Figure 5 by molecular docking. Probably due to the presence of two -NH_3_ ligands, DDP preferred to bind on the side of 18-crown-6. At the same time, from the top view, DDP was not at the median position of 18-crown-6, but was slightly off. This fitted structure is also shown with the bond length of hydrogen bonding. Calculations revealed a binding energy (Δ*E*) of −2.63 kcal/mol, indicating a robust electrostatic-driven hydrogen bonding interaction [40]. The abovementioned UV–visible spectroscopy revealed that the hydrogen bond network between the two compounds enhanced the solubility of DDP to some extent.

Analysis of DDP’s molecular structure elucidated its low aqueous solubility. The spatial structure and chemical properties were analyzed, and two neutral molecules -NH_3_ and two anions Cl^−^ cooperated with Pt to form a square planar framework structure, the DDP. Essentially, both ligands, -NH_3_ and Cl^−^, possessed good water solubility on their own; however, the hydrophilic contribution of -NH_3_ and Cl^−^ was diminished upon the formation of a stabilized framework with Pt. The burial of the hydrophilic nature within the DDP molecule resulted in its poor aqueous solubility [41]. Additionally, the semi-symmetric structure of the DDP molecule rendered it a less polar substance, whereas water is polar solvent, which might contribute to its lower solubility [42].

The introduction of 18-crown-6 improved the solubility of DDP, likely due to the formation of the DDP@18-crown-6 complex, which altered the molecular configuration, chemical properties, and polarity. Specifically, 18-crown-6, inherently more soluble in water, interacted with -NH_3_ ligands of DDP through hydrogen bonding, thereby increasing the inclusion of hydrophilic groups in the complex. The spatial structure of the DDP@18-crown-6 complex revealed an increased proportion of hydrophilic groups, effectively mitigating the effects of solubility reduction of ligands due to the expanded molecular framework. This increase in the proportion of hydrophilic groups enhanced the overall water solubility of the molecule to a large extent [43]. From a polarity perspective, the formation of a complex between 18-crown-6 and DDP increased the polarity of the whole system, thus dissolving more readily and exhibiting improved solubility in water [42,44]. Therefore, the formation of the DDP@18-crown-6 complex substantially was expected to enhance DDP’s solubility in aqueous environments.

Taken together, the FT-IR and Raman spectra and molecular docking elucidated that DDP and 18-crown-6 could combine in ultrapure aqueous solvents via a hydrogen bonding network and form the DDP@18-crown-6 complex.

### 2.6. Phase Solubility Study

As shown in the phase solubility diagrams (Figure 6), the solubility of DDP increased linearly with the concentration of 18-crown-6, revealing an A_L_-type (linear) phase diagram and confirming the formation of a 1:1 complex between DDP and 18-crown-6 in aqueous solution [31]. In a 640 mM (maximum) solution of 18-crown-6, the solubility of DDP improved by a factor of 5.03. The intrinsic solubility (*S*_0_) of DDP was measured as 1.540 mg/mL (5.169 mM), closely aligning with the reported solubility of 1 mg/mL [45]. Calculations using Equation (1) yielded a complexation constant (*K*_1:1_) of 6.295 M^−1^, and using Equation (2) determined the *p* value for DDP in 640 mM 18-crown-6 solution to be 4.035, indicating that the majority of DDP dissolved in water due to complex formation. We imagined that the solubilization effects of 18-crown-6 could be controlled by tuning the concentration. Besides, the elevation of solubility could facilitate the future design and development of high-dose formulations.

### 2.7. Stoichiometry by Job’s Plot

Job’s plot, or the continuous variation method, was employed to elucidate the stoichiometry of the complex [46]. In a solution containing host and guest species, the guest can bind to the host. By varying their ratio while maintaining a constant total concentration, the system achieves optimal UV or fluorescence absorption when the guest and host are stoichiometrically matched. UV–visible spectrophotometric analysis plotted absorbance against *X*_m_ of DDP in the DDP-18-crown-6 complex (Figure 7A). The absorbance peaked at a molar fraction of 0.5 (*p* < 0.0001), indicating an optimal 1:1 binding ratio between DDP and 18-crown-6, consistent with molecular docking and phase solubility studies. Additionally, Job’s plot revealed a minor peak at a molar fraction of 0.1 (i.e., DDP:18-crown-6 = 1:9), suggesting the possible presence of a cyclic inclusion complex formed by the physical adsorption of DDP with multiple 18-crown-6 molecules (Figure 7B) [17], though this was not the predominant structure. In this plausible structure, we assumed that six 18-crown-6 molecules each form a hydrogen bond binding to -NH_3_ on DDP (blue dashed line), while the other three 18-crown-6 molecules form a weak electrostatic interaction with Pt (red dashed line). In contrast, the optimal DDP@18-crown-6 complex (*X*_m_ = 0.5) lacked electrostatic interactions, likely due to steric hindrance from the two -NH_3_ ligands, which obstructed the interaction between Pt and the oxygen atoms of 18-crown-6. Instead, the -NH_3_ groups preferentially formed hydrogen bonds with the oxygen atoms of 18-crown-6. Conversely, the non-optimal conformation (*X*_m_ = 0.9) featured six 18-crown-6 molecules that entirely encapsulate the -NH_3_ ligands, thereby exposing the Pt binding sites and creating spatial opportunities for 18-crown-6 binding.

Thus far, the solubilization effects of 18-crown-6 on DDP were extremely significant, with a five-fold increase in a 640 mM (maximum) solution of 18-crown-6. This is due to the formation of the DDP@18-crown-6 complex with better water solubility. According to molecular docking, phase solubility studies, and Job’s analysis, one molecule of DDP and one molecule of 18-crown-6 were bonded via a hydrogen bonding network to form the optimal structural formula of the DDP@18-crown-6 complex.

### 2.8. Stability Analysis of DDP@18-Crown-6 Complex

Focusing on the issue of DDP stability, we investigated the effect of the formation of the DDP@18-crown-6 complex on the time-evolution stability and photostability of DDP.

#### 2.8.1. Time-Evolution Stability Studies of the Complex

The addition of 18-crown-6 was employed to enhance not only the solubility, but also the stability of DDP in aqueous solutions. During storage, the Cl^−^ ligands of DDP in pure water are readily replaced by solvent water molecules, leading to the formation of various hydrolysis products such as [PtCl(H_2_O)(NH_3_)_2_]^+^ and [Pt(H_2_O)_2_(NH_3_)_2_]. This hydrolysis results in diminished or even complete loss of activity [7,8]. To mitigate hydrolysis, a 0.9% NaCl solution is sometimes employed to enhance Cl^−^ competition and hinder water molecule substitution. However, this method only maintains activity transiently [47,48]. Therefore, 18-crown-6 is introduced to enhance the stability of DDP further. According to molecular docking analysis, when 18-crown-6 binds to DDP, it could increase the spatial site resistance of water molecule ligand binding, thus inhibiting the generation of [PtCl(H_2_O)(NH_3_)_2_]^+^, [Pt(H_2_O)_2_(NH_3_)_2_], and hydroxo-bridged dimer [49]. The decrease in hydrolysis capacity increased the stability of DDP in water. Additionally, from an electronic donor–acceptor perspective, the greater electronegativity of oxygen compared to nitrogen caused the hydrogen bond network to shift electron density towards the oxygen atoms, reducing the electron cloud density near the nitrogen atoms [50]. To enhance system stability, the Pt atom in DDP attracted electrons from Cl^−^ to compensate for the electron deficiency around the -NH_3_ ligands. This adjustment in electron density strengthened the Pt-Cl bond, thereby reducing the formation of hydrolysis products and increasing the stability of DDP.

As illustrated in Figure 8A, after 4 days, the concentration of DDP in the DDP@18-crown-6 complex solution remained at 99.08%, surpassing that in pure water (81.03%) (*p* < 0.0001) and 0.9% NaCl solutions (97.70%) (*p* = 0.0035). The DDP@18-crown-6 complex in 0.9% NaCl solutions exhibited the highest stability, with DDP concentration reaching 99.52% after 4 days (*p* = 0.0069), indicating optimal stability due to the combined effects of NaCl and 18-crown-6. The enhanced stability was likely due to the dual protection of Cl^−^ competition and complex formation. Additionally, varying concentrations of 18-crown-6 improved DDP stability to a similar degree (Figure 8B), with minimal variation across different concentrations. Therefore, we did not need to consume a large amount of 18-crown-6 for stabilization.

#### 2.8.2. Photostability Studies of the Complex

The poor photostability of DDP significantly impairs its therapeutic efficacy. Under light exposure, DDP is prone to photohydrolysis and photodegradation [9]. Photohydrolysis leads to the substitution of Cl^−^ ligands in DDP, forming aqueous complexes and reducing its activity [49,51]. Additionally, photodegradation, particularly the cleavage of the light-sensitive Pt-Cl bond, can cause DDP breakdown and Pt precipitation [52]. To address these issues, 18-crown-6 was employed. Similar to the time-evolution stabilization mechanisms, the DDP@18-crown-6 complex combined with 0.9% NaCl enhanced the Pt-Cl bond through steric hindrance, altered electron density, and Cl^−^ competition in the photostability studies, whereas the augmentation of the Pt-Cl bond reduced the photohydrolysis and photodegradation reactions of DDP [47,48]. We predict that this dual action will also mitigate photohydrolysis and photodegradation of DDP.

To corroborate this conjecture, the impact of 18-crown-6 on the photostability of DDP was assessed. As depicted in Figure 9A, after 45 min of exposure to light, the degradation rate of DDP in the 18-crown-6 complex solution (2.49%) was substantially lower compared to that in pure water (7.55%) (*p* < 0.0001) and 0.9% NaCl solutions (4.31%) (*p* < 0.0001), with DDP content remaining at 97.51%. Furthermore, the presence of 0.9% NaCl further enhanced the photostability of DDP. Specifically, the DDP@18-crown-6 complex in 0.9% NaCl maintained 99.09% of DDP after 45 min of light exposure (*p* < 0.0001), indicating optimal photostability. Therefore, the addition of the two also had a combined photostabilization effect. Additionally, various concentrations of 18-crown-6 comparably improved the photostability of DDP, with minimal variation observed (Figure 9B). Hence, a low dose of 18-crown-6 will suffice for stabilization.

The experimental results confirmed that the formation of the DDP@18-crown-6 complex significantly enhanced the time-evolution stability and photostability of DDP. Characterization data suggested that this stability enhancement was likely mediated by steric hindrance and electronic induction effects. Furthermore, it was indicated that the combined action of Cl^−^ competition facilitated by 0.9% NaCl and the steric and electronic effects of the complex synergistically improved DDP stability. Hereto, we have provided a strong basis for considering the synergistic use of 0.9% NaCl and 18-crown-6 in the development of DDP formulations in order to arrive at optimal solubility and stability.

### 2.9. Evaluation of Antitumor Activity Against Lung Cancer

As a prototypical antitumor agent, DDP exerts its therapeutic effects by inducing cellular damage, which can be quantitatively assessed through cell viability assays [53]. This study employed the CCK-8 assay to evaluate the impact of varying concentrations of DDP and the DDP@18-crown-6 complex on A549 cell viability, and to determine their respective IC_50_ values. Figure 10A revealed that both DDP and the DDP@18-crown-6 complex exhibited dose-dependent cytotoxicity against A549 cells. The cytotoxicity of both of them was already significantly different from a concentration of 2 µg/mL (*p* < 0.001), indicating that the formation of the complexes enhanced the antitumor activity of the drugs more significantly. Notably, the DDP@18-crown-6 complex demonstrated enhanced potency, with an IC_50_ of 5.043 µg/mL compared to 8.629 µg/mL for pristine DDP. The results might be interpreted in two measures: (1) The cytotoxicity of DDP was enhanced after being included in 18-crown-6; (2) 18-crown-6 per se was toxic and added to the cytotoxicity.

To confirm the non-toxic nature of 18-crown-6 and preclude the influence of it, its impact on A549 viability was also assessed using the CCK-8 assay, ensuring the compound’s safety in the tumoral context. Experimental data (Figure 10B) demonstrated that A549 cell viability remained above 98% even at 74 µg/mL of 18-crown-6, indicating no significant cytotoxicity of 18-crown-6. Thus, the enhanced antitumor activity of the DDP@18-crown-6 complex compared to free DDP was not attributed to the toxicity of 18-crown-6, but rather to the complex formation. This complex improved the solubility and stability of DDP, ensuring that the drug effectively reached the cancer cells. In contrast, the physical precipitation and/or chemical degradation of free DDP led to loss of potency before it could exert its therapeutic effects. It was inferred that the rise in anticancer efficacy stemmed from elevated solubility and stability of DDP. As an implication, the proposed DDP@18-crown-6 complex might exhibit better clinical outcomes compared to conventional formulations.

## 3. Materials and Methods

### 3.1. Materials

All chemicals employed in this study were of reagent or analytical grade, sourced from reputable commercial suppliers. DDP was obtained from Solarbio Science and Technology Co., Ltd. (Beijing, China), and 18-crown-6 and NaCl were supplied by Shanghai Macklin Biochemical Technology Co., Ltd. (Shanghai, China). PEG400 was acquired from Selleck (Shanghai, China), and the Cell Counting Kit-8 (CCK-8) was provided by Guangzhou Dibo Biotechnology Co., Ltd. (Guangzhou, China).

### 3.2. Preparation of DDP@18-Crown-6 Complex

Equimolar ratios of DDP and 18-crown-6 were dissolved in 4 mL of ultrapure water and subjected to magnetic stirring at room temperature for a suitable time using a certain rotation rate. The resulting mixture was filtered through a Büchner funnel to eliminate insoluble residues, yielding a clear solution of the DDP@18-crown-6 complex [54].

#### 3.2.1. Optimization of Preparation Time

Equimolar ratios of DDP and 18-crown-6 were dissolved in 4 mL of ultrapure water. The solutions were then magnetically stirred at 800 rpm for 12, 24, 36, 48, 60, and 72 h at room temperature. The resulting DDP@18-crown-6 complexes were isolated by filtration through a Büchner funnel to remove any insoluble residues, yielding clear solutions. Absorbance at 301 nm was subsequently assessed using a UV–visible spectrophotometer (V730, JASCO, Tokyo, Japan) to evaluate and optimize the complex preparation time.

#### 3.2.2. Optimization of Rotation Rate

An equimolar ratio of DDP and 18-crown-6 were dissolved in 4 mL of ultrapure water. The mixtures were subjected to magnetic stirring at room temperature at varying rates of 600, 800, 1000, 1200, and 1400 rpm for 48 h. After stirring, the DDP@18-crown-6 complexes were isolated by filtration through a Büchner funnel to remove insoluble DDP, yielding clear solutions. The absorbance of this solution at 301 nm was measured with a UV–visible spectrophotometer to assess and optimize the complex preparation rate.

### 3.3. Characterization of DDP@18-Crown-6 Complex

#### 3.3.1. UV–Visible Spectrophotometric Studies

The DDP@18-crown-6 complex was prepared using the optimized protocol. Solutions of DDP and 18-crown-6 were prepared and filtered to obtain clear solutions. UV–visible spectrophotometry was employed for full-wavelength scanning to assess the UV absorbance of DDP, 18-crown-6, and the DDP@18-crown-6 complex, focusing on changes in the absorbance peak at 301 nm.

#### 3.3.2. FT-IR Spectroscopic Studies

The DDP@18-crown-6 complex was prepared using the optimized method. Following this, DDP, 18-crown-6, and the DDP@18-crown-6 complex were dried to obtain powders. FT-IR spectroscopy (Spectrum Two, PerkinElmer, Waltham, MA, USA) was performed at a resolution of 4 cm^−1^ over the range of 4000 to 400 cm^−1^ to analyze the interaction sites between DDP and 18-crown-6.

#### 3.3.3. Raman Spectroscopic Studies

The DDP@18-crown-6 complex was synthesized according to the optimized protocol. Subsequently, DDP, 18-crown-6, and DDP@18-crown-6 complex were dried to obtain powders. Raman spectra of the samples were acquired using confocal in situ Raman spectroscopy (LabRAM HR Evolution, Horiba Scientific, Kyoto, Japan), covering a range from 200 cm^−1^ to 2000 cm^−1^ with excitation at 532 nm [55]. All spectra were recorded at room temperature (20 °C) in the solid phase.

#### 3.3.4. Molecular Docking

Initially, the structures of DDP and 18-crown-6 were constructed. Subsequently, DDP was used as the ligand, and 18-crown-6 as the receptor for molecular docking using AutoDock 4.2, with 200 docking simulations performed. The docking results were clustered to identify the optimal cluster, from which the best binding conformation was selected. Finally, interaction analysis of the DDP and 18-crown-6 was conducted to predict binding modes and affinity. The specific measures were as follows:

In this study, AutoDock 4.2, employing the Lamarckian genetic algorithm, was used for docking simulations. Structural data for DDP and 18-crown-6 were obtained from the PubChem database and prepared using AutoDockTools, which involved adding hydrogen atoms, applying Gasteiger–Hückel partial charges, merging non-polar hydrogens, and setting rotatable bonds. Specifically, σ-bonds between heavy atoms in the DDP structure were treated as rotatable, while 18-crown-6 was considered rigid. The docking procedure involved positioning a 25 × 25 × 25 Å^3^ grid box (with a step size of 0.375 Å) around the central site of 18-crown-6, with DDP undergoing 200 independent docking runs within this grid.

During the docking process, the Lamarckian genetic algorithm generated 150 random orientations and conformations for DDP, with each generation undergoing up to 1,500,000 energy optimizations. The top 10 conformations were selected for further generations, with a gene crossover rate of 0.8 and a mutation rate of 0.02, continuing for 27,000 generations until convergence. All other parameters were set to default values in AutoDock 4.2. Post-docking, the 200 results were clustered, and the optimal binding conformation was chosen based on the lowest docking score from the best cluster. The selected conformation was then analyzed for molecular interactions, with key parameters such as bond lengths and angles extracted and visualized using PyMOL (PyMOL Molecular Graphics System, Version 1.7.4 Schrödinger, LLC, New York, NY, USA) to create a three-dimensional interaction diagram.

### 3.4. Phase Solubility Study

Solubility analysis was conducted following the method of Higuchi and Conners [31,38,56]. Excess DDP was added to 3 mL of aqueous 18-crown-6 solutions at varying concentrations (0, 40, 80, 160, 320, and 640 mM). Six samples were stirred at 1200 rpm at room temperature for 48 h. After stirring, the samples were filtered through a Büchner funnel to collect the clear solutions, yielding the DDP@18-crown-6 complex. The absorbance at 301 nm was then measured using a UV–visible spectrophotometer to determine the solubility of DDP in solutions of different 18-crown-6 concentrations (Appendix A).

The complexation constant (*K*_1:1_) was calculated under the assumption of a 1:1 binding ratio of DDP to 18-crown-6 using Equation (1).
(1)$$K_{1:1}=slope/S_{0}\left(1-slpoe\right)$$

Here, *K*_1:1_ represents the complexation constant. *S*_0_ denotes the intrinsic solubility of DDP, which was determined from the solubility of DDP in water without 18-crown-6. The slope is derived from the plot of drug concentration versus 18-crown-6 concentration.

Additionally, the partition coefficient (*P*) of the DDP complex, representing the ratio of drug concentration in the complex to that in water, was calculated using Equation (2) [57].
(2)$$P=S_{tot}-S_{0}/S_{0}$$

In the equations, *S_tot_* is the solubility of DDP in the presence of a specific concentration of 18-crown-6.

### 3.5. Stoichiometry by Job’s Plot

The optimal binding ratio of DDP to 18-crown-6 was determined using the Job’s plot [58,59]. Equal molar solutions of DDP and 18-crown-6 were prepared. The total molar concentration of DDP and 18-crown-6 was fixed, and the molar fractions of DDP to 18-crown-6 in the mixtures ranged from 1:9 to 8:2 (by volume). Absorbance at 301 nm for these mixtures was measured using a UV–visible spectrophotometer. The difference in absorbance values was plotted against the molar fraction of DDP (*n*_DDP_/*n*_total_, Equation (3)). The inflection points of the resulting curve indicated the optimal binding ratio. The 1:1 ratio was confirmed when the Job’s plot exhibited a maximum at molar fraction (*X*_m_) = 0.5.
(3)$$X\left(\mathrm{Mole}\;\mathrm{fraction}\right)=\left[\mathrm{DDP}\right]/\left(\left[\mathrm{DDP}\right]+\left.\left[18-\mathrm{crown}\;\mathrm{ether}-6\right]\right)\right.$$
where [DDP] is the molar concentration of DDP, and [18-crown-6] is the molar concentration of 18-crown-6.

### 3.6. Stability of DDP@18-Crown-6 Complex

#### 3.6.1. Time-Evolution Stability Studies of the Complex

DDP powders (1.5 g) were dissolved in equal volumes of ultrapure water, 0.9% NaCl solution, equimolar 18-crown-6 solution, and equimolar 18-crown-6 in 0.9% NaCl solution. Each solution was stirred magnetically at 1200 rpm for 48 h. The mixtures were then filtered through a Büchner funnel to remove undissolved DDP, resulting in four clear solutions: DDP (in H_2_O), DDP (in 0.9% NaCl aqueous solution), DDP@18-crown-6 complex (in H_2_O), and DDP@18-crown-6 complex (in 0.9% NaCl aqueous solution). These solutions were stored at room temperature, and UV absorbance at 301 nm was measured at 0, 1, 2, and 3 days. The concentration of DDP at different storage times was calculated, and the percentage of DDP was calculated according to Equation (4). Notably, the entire experimental procedure was completely protected from light.
(4)$$\mathrm{Percentage}\;\mathrm{of}\;\mathrm{DDP}\;\left(\%\right)=C\left(\mathrm{DDP}\;\mathrm{at}\;0\;\mathrm{day}\right)/C\left(\mathrm{DDP}\;\mathrm{at}\;\mathrm{specific}\;\mathrm{time}\right)$$
where *C*(DDP at 0 day) is the concentration of DDP in the solution at day 0, and *C*(DDP at specific day) is the concentration of DDP in the solution at a specific day.

The effect of varying concentrations of 18-crown-6 on the stability of DDP over time was investigated. DDP powders (1.5 g) were dissolved in equal volumes of 18-crown-6 solutions at concentrations of 5 mM, 40 mM, 80 mM, and 160 mM. Each solution was stirred magnetically at 1200 rpm for 48 h. The mixtures were then filtered through a Büchner funnel to remove insoluble DDP, yielding clear solutions. These solutions were stored at room temperature away from light, and UV absorbance at 301 nm was measured at 0, 1, 2, and 3 days. The concentration of DDP at different storage times was calculated, and the percentage of DDP was calculated according to Equation (4).

#### 3.6.2. Photostability Studies of the Complex

DDP powders (1.5 g) were dissolved in equal volumes of ultrapure water, 0.9% NaCl solution, equimolar 18-crown-6 solution, and equimolar 18-crown-6 in 0.9% NaCl solution. Each solution was stirred magnetically at 1200 rpm for 48 h. Following this, the samples were filtered through a Büchner funnel to remove undissolved DDP, resulting in four clear solutions: DDP (in H_2_O), DDP (in 0.9% NaCl aqueous solution), DDP@18-crown-6 complex (in H_2_O), and DDP@18-crown-6 complex (in 0.9% NaCl aqueous solution). These solutions were exposed to light (4500 ± 500 lux), and UV absorbance at 301 nm was measured at 0, 15, 30, and 45 min. The concentration of DDP at different illumination times was calculated, and the percentage of DDP was calculated according to Equation (5).
(5)$$\mathrm{Percentage}\;\mathrm{of}\;\mathrm{DDP}\;\left(\%\right)=C\left(\mathrm{DDP}\;\mathrm{at}\;0\;\min\right)/C\left(\mathrm{DDP}\;\mathrm{at}\;\mathrm{specific}\;\min\right)$$
where *C*(DDP at 0 min) is the concentration of DDP in the solution at 0 min, and *C*(DDP at specific min) is the concentration of DDP in the solution at a specific minute.

The impact of varying concentrations of 18-crown-6 on the photostability of DDP was investigated. DDP powders (1.5 g) were dissolved in equal volumes of 18-crown-6 solutions at concentrations of 5 mM, 40 mM, 80 mM, and 160 mM. These solutions were magnetically stirred at 1200 rpm for 48 h. Following filtration through a Büchner funnel to remove undissolved DDP, clear solutions were obtained. These solutions were exposed to light (4500 ± 500 lux), and UV absorbance at 301 nm was measured at intervals of 0, 15, 30, and 45 min. The concentration of DDP at different illumination times was calculated, and the percentage of DDP was calculated according to Equation (5).

### 3.7. Evaluation of Antitumor Activity Against Lung Cancer

#### 3.7.1. Cell Culture Studies

A549 cells, sourced from the American Type Culture Collection (ATCC CCL-185, Rockville, MD, USA), were cultured in tissue-treated T-25 flasks using 0.25% EDTA-trypsin Dulbecco’s Modified Eagle Medium (DMEM, NEST Biotechnology Co., Ltd., Wuxi, China), enriched with 10% fetal bovine serum (Guangzhou Dibo Biotechnology, Guangzhou, China) and 1% penicillin–streptomycin. The cultures were maintained at 37 °C in a 5% CO_2_ atmosphere.

#### 3.7.2. Cell Viability Assays

The cytotoxicity of DDP and the DDP@18-crown-6 complex in A549 cells was assessed using the CCK-8 assay. In brief, 1 × 10^4^ cells per well were seeded into 96-well plates and cultured overnight at 37 °C in 5% CO_2_. On the following day, cells were exposed to varying concentrations (2–32 μg/mL) of DDP and DDP@18-crown-6 complex for 24 h. Post-treatment, 1% CCK-8 reagent in DMEM was added to each well, and the plates were incubated under the same conditions. Absorbance at 450 nm was measured using a microplate reader (Epoch 2, Berton Instruments, Inc., Winooski, VT, USA) to evaluate cell viability. The percentage of cell viability was calculated by comparing the absorbance of treated samples to that of the untreated control group. IC_50_ values for each treatment were determined by logarithmic transformation and non-linear curve fitting using GraphPad Prism version 9.5 (GraphPad, Inc., San Diego, CA, USA). Additionally, the cytotoxicity of 18-crown-6 (1.156–74 μg/mL) was evaluated using the same method to confirm its safety.

### 3.8. Statistical Analysis

All data are presented as mean ± standard deviation (SD). All experiments were conducted in triplicate. Statistical analysis was performed using GraphPad Prism 9.5 (GraphPad Software, San Diego, CA, USA) and Origin 2017 (OriginLab Corporation, Northampton, MA, USA). Significance between groups was assessed using the *t*-test and one-way ANOVA, and the difference was considered to be significant when *p* < 0.05. Significance levels were denoted as *, **, ***, and **** for *p* < 0.05, *p* < 0.01, *p* < 0.001, and *p* < 0.0001, respectively.

## 4. Conclusions

DDP is a classical antitumor agent used in cancer therapy. However, its clinical utility is hindered by poor solubility and instability. To address these limitations, we utilized the macrocyclic host molecule 18-crown-6, which enhanced the solubility and stability of DDP through host–guest supramolecular chemistry, while boosting its therapeutic efficacy. In the presence of 18-crown-6 (640 mM), the solubility of DDP in aqueous systems was increased more than five-fold. This might be due to the forming of a water-soluble DDP@18-crown-6 complex via a hydrogen bonding network between the -NH_3_ groups of DDP and the oxygen atoms of 18-crown-6. The formation of the DDP@18-crown-6 complex significantly enhanced both the storage stability and photostability of DDP. Notably, the presence of 18-crown-6 in conjunction with 0.9% NaCl further improved the stability of DDP, likely due to the dual protective effect of ion competition and complex formation. In vitro antitumor assays revealed that the DDP@18-crown-6 complex (IC_50_ = 5.043 µg/mL) exhibited superior antitumor activity compared to free DDP (IC_50_ = 8.629 µg/mL). This demonstrated that complexation did not diminish, but rather enhanced, the efficacy of DDP owing to improved solubility and stability. Consequently, the use of 18-crown-6 offered significant potential to optimize the clinical application and development prospects of DDP.

The promising results of this study suggest a new avenue for drug research and development. 18-Crown-6, which significantly enhanced both solubility and stability of DDP, could serve as a platform for optimizing other drug molecules containing -NH_3_ groups or metal elements (such as Pt, silver, lithium, vanadium, and gold) [60]. Supramolecular chemistry offers a strategy for refining these drugs, potentially benefiting the development of therapies for cancer and various diseases. Moreover, modifications using this approach do not compromise therapeutic efficacy; rather, they may enhance the anticancer activity of DDP. Thus, in future studies, supramolecular chemistry could play a pivotal role in augmenting drug efficacy. Overall, 18-crown-6, as a supramolecular platform, presented a promising method for modifying drug solubility, stability, and activity.

## Figures and Tables

**Figure 1 ijms-25-13411-f001:**
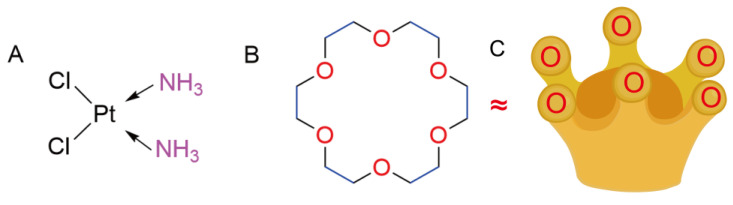
Chemical structure of cisplatin (**A**) and 18-crown-6 (**B**), and “crown” shaped like 18-crown-6 (**C**).

**Figure 2 ijms-25-13411-f002:**
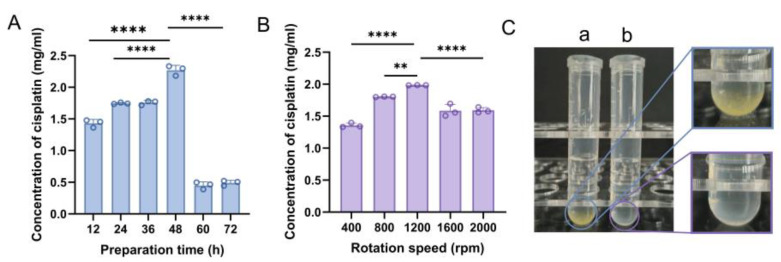
The optimization of preparation time (**A**) and rotation rate (**B**) of DDP@18-crown-6 complex (*n* = 3) and optical photograph of cisplatin dispersed or dissolved (**C**) in ultrapure water (a) or 18-crown-6 aqueous solution (b). ANOVA or *t* tests were utilized to determine whether there are significant differences between the data. *p* value style: ** *p* < 0.01; **** *p* < 0.0001.

**Figure 3 ijms-25-13411-f003:**
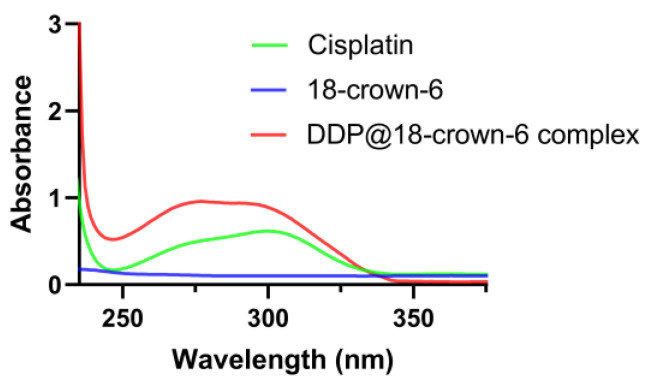
The UV–visible spectrogram of cisplatin, 18-crown-6, and the DDP@18-crown-6 complex.

**Figure 4 ijms-25-13411-f004:**
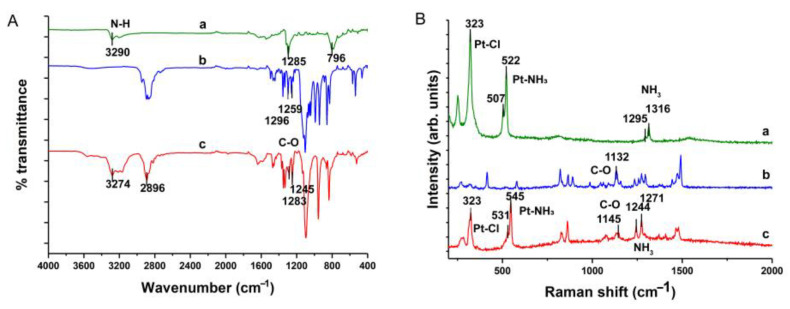
The FT-IR spectrogram (**A**) and Raman spectrogram (**B**) of cisplatin (a), 18-crown-6 (b), and DDP@18-crown-6 complex (c).

**Figure 5 ijms-25-13411-f005:**
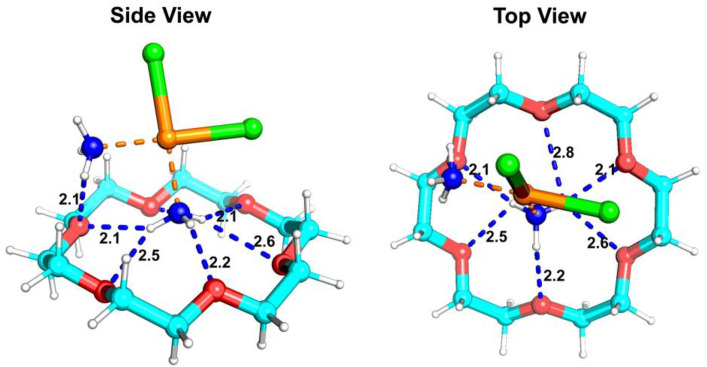
The optimal binding conformation of cisplatin to 18-crown-6. The C, N, O, H, Cl, and Pt atoms are colored in cyan, blue, red, white, green, and orange, respectively. Orange dashed lines are coordinate covalent bonds, and blue dashed lines are hydrogen bonds.

**Figure 6 ijms-25-13411-f006:**
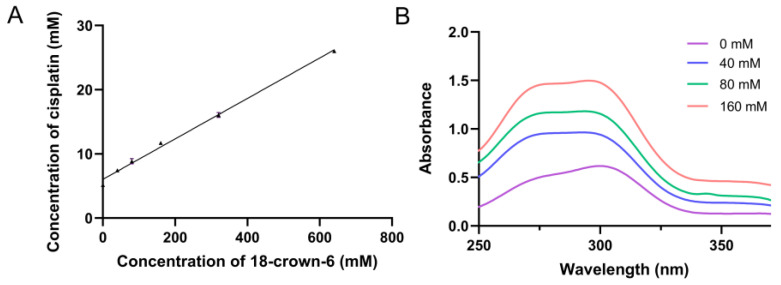
Phase solubility diagram (**A**) for cisplatin with 18-crown-6 in aqueous solution (*n* = 3) and UV–visible spectrogram of the complex at different 18-crown-6 concentrations (**B**). The linear regression equation for the phase solubility curves was *Y* = 0.03151*X* + 6.036 (**A**) and the correlation coefficient, *R*^2^, was greater than 0.99.

**Figure 7 ijms-25-13411-f007:**
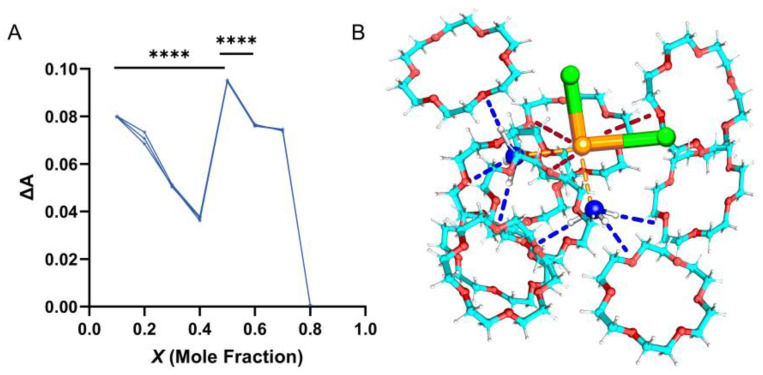
Job’s plot (**A**) (*n* = 3) and binding structural schematic (**B**) of cisplatin and 18-crown-6 at *X*_m_ = 0.9. The C, N, O, H, Cl, and Pt atoms are colored in cyan, blue, red, white, green, and orange, respectively. Orange dashed lines are coordinate covalent bonds. Red dashed lines indicate electrostatic interaction, and blue dashed lines are hydrogen bonds. ANOVA or *t* tests were utilized to determine whether there are significant differences between the data. *p* value style: **** *p* < 0.0001.

**Figure 8 ijms-25-13411-f008:**
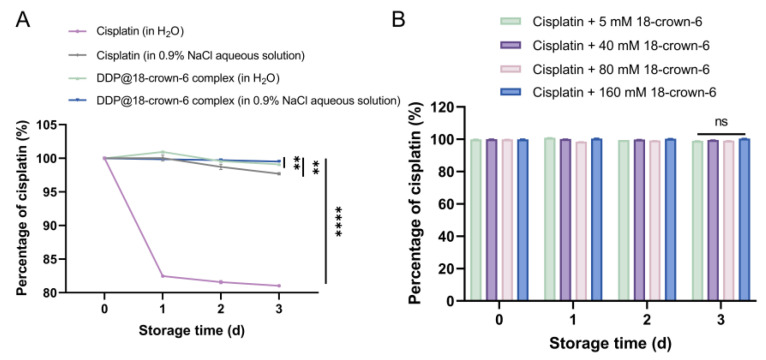
(**A**) Time-evolution stability of cisplatin (in H_2_O), cisplatin (in 0.9% NaCl aqueous solution), DDP@18-crown-6 complex (in H_2_O), and DDP@18-crown-6 complex (0.9% in NaCl aqueous solution); and (**B**) time-evolution stability of cisplatin at different 18-crown-6 concentrations (*n* = 3). ANOVA or *t* tests were utilized to determine whether there are significant differences between the data. *p* value style: ns: not significant (*p* > 0.05); ** *p* < 0.01; **** *p* < 0.0001.

**Figure 9 ijms-25-13411-f009:**
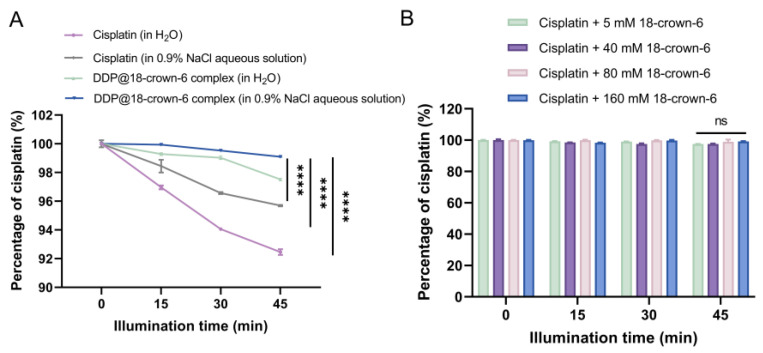
(**A**) Photostability of cisplatin (in H_2_O), cisplatin (in 0.9% NaCl aqueous solution), DDP@18-crown-6 complex (in H_2_O), and DDP@18-crown-6 complex (in 0.9% NaCl aqueous solution); and (**B**) photostability of cisplatin at different 18-crown-6 concentrations (*n* = 3). ANOVA or *t* tests were utilized to determine whether there are significant differences between the data. *p* value style: ns: not significant (*p* > 0.05); **** *p* < 0.0001.

**Figure 10 ijms-25-13411-f010:**
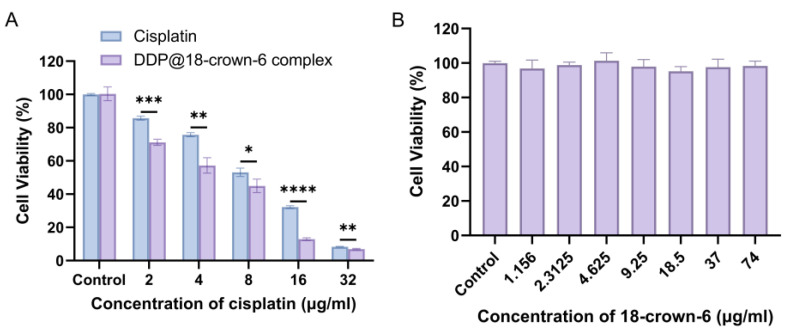
In vitro cytotoxicity of cisplatin and DDP@18-crown-6 complex (**A**), and 18-crown-6 (**B**). (*n* = 3) ANOVA or *t* tests were utilized to determine whether there are significant differences between the data. *p* value style: * *p* < 0.05; ** *p* < 0.01; *** *p* < 0.001; **** *p* < 0.0001.

## Data Availability

All data generated or analyzed during this study are included in this published article and its Appendix A files.

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
