# Peer review of "Construction of Cisplatin-18-Crown-6 Complexes Through Supramolecular Chemistry to Improve Solubility, Stability, and Antitumor Activity"

_ijms, 2024, doi:10.3390/ijms252413411_

Round 1

Reviewer 1 Report

Comments and Suggestions for Authors

In this article, the authors utilized 18-crown-6 as a host molecule to solubilize and stabilize cisplatin, addressing the limitations of poor solubility and stability that hinder the therapeutic effectiveness of cisplatin in cancer treatment. This is a novel and interesting study, and the authors employed various analytical methods to demonstrate that the DDP@18-crown-6 complex exhibits improved solubility, stability, and tumor-killing effects. Overall, this work provides robust data and reliable conclusions. However, one issue requires further consideration:

In Figure 10, the DDP@18-crown-6 complex shows stronger cytotoxicity compared to cisplatin alone. Could this increased cytotoxicity be attributed to residual solvents or compounds introduced during the complexation process between DDP and 18-crown-6?

Author Response

Review 1:

In this article, the authors utilized 18-crown-6 as a host molecule to solubilize and stabilize cisplatin, addressing the limitations of poor solubility and stability that hinder the therapeutic effectiveness of cisplatin in cancer treatment. This is a novel and interesting study, and the authors employed various analytical methods to demonstrate that the DDP@18-crown-6 complex exhibits improved solubility, stability, and tumor-killing effects. Overall, this work provides robust data and reliable conclusions. However, one issue requires further consideration:

In Figure 10, the DDP@18-crown-6 complex shows stronger cytotoxicity compared to cisplatin alone. Could this increased cytotoxicity be attributed to residual solvents or compounds introduced during the complexation process between DDP and 18-crown-6?

Response: Thank you for your valuable comments.

(1) Firstly, the increased toxicity of the DDP@18-crown-6 complex is unrelated to the residual solvent introduced. As detailed in Section 4.2, the preparation of the DDP@18-crown-6 complex involves the use of ultrapure water as the solvent, with no other hazardous substances involved. Therefore, compared to cisplatin alone, the observed increase of toxicity in the complex is not influenced by the solvent.

(2) Furthermore, we assessed the toxicity of 18-crown-6 aqueous solution at equivalent concentrations, as described in Section 2.9. The results demonstrated that concentrations ranging from 0 to 74 µg/mL of 18-crown-6 did not exhibit any toxicity to A549 cells, ruling out the potential impact of the compound on the toxicity of DDP@18-crown-6 complex.

(3) Additionally, no chemical reactions occur during the complex formation, ensuring that no new compounds are introduced, thus preventing any contribution to the toxicity.

In conclusion, the increased toxicity of the DDP@18-crown-6 complex, compared to cisplatin alone, can be attributed solely to the formation of the complex, rather than residual solvent or the introduction of other compounds.

Reviewer 2 Report

Comments and Suggestions for Authors

The authors of the presented study analyzed the possibility of forming complexes between cisplatin, a popular and widely used anticancer drug, and 18-crown-6, aiming to significantly improve its water solubility, enhance its stability, and potentially influence its pharmacological activity. The proposed work is thoughtfully constructed and comprehensively addresses the researched issue, incorporating various spectroscopic techniques as well as in vitro studies to evaluate the biological activity of the proposed complexes. I recommend considering the publication of this work after addressing certain comments that arise from my review of the manuscript.

1.      The authors frequently refer to statistical testing at various significance levels and indicate statistical differences at specific levels of significance using asterisks. However, this became clear to me only after reading the methodology section. In my opinion, appropriate annotations regarding the testing and statistical significance of observed differences should be clarified in the sections where the data are presented, for instance, in figures 2, 7, or 8, especially since the article's structure, as proposed by this publisher, places the methodology section after the presentation of results.

2.      Figure 2 is challenging to assess from a substantive perspective due to the lack of information regarding the cisplatin solution concentration and the absence of a definition for the cisplatin complex with the complexing agent. There are three data series presented, but no clear relationships between the concentrations of the described systems can be discerned. How can the increase in the absorbance of the complex relative to pure cisplatin be interpreted if there is no clarity on the systems that were actually studied? Detailed information about the presented systems must be included in the figure description.

Author Response

Review 2:

The authors of the presented study analyzed the possibility of forming complexes between cisplatin, a popular and widely used anticancer drug, and 18-crown-6, aiming to significantly improve its water solubility, enhance its stability, and potentially influence its pharmacological activity. The proposed work is thoughtfully constructed and comprehensively addresses the researched issue, incorporating various spectroscopic techniques as well as in vitro studies to evaluate the biological activity of the proposed complexes. I recommend considering the publication of this work after addressing certain comments that arise from my review of the manuscript.

Q1. The authors frequently refer to statistical testing at various significance levels and indicate statistical differences at specific levels of significance using asterisks. However, this became clear to me only after reading the methodology section. In my opinion, appropriate annotations regarding the testing and statistical significance of observed differences should be clarified in the sections where the data are presented, for instance, in figures 2, 7, or 8, especially since the article's structure, as proposed by this publisher, places the methodology section after the presentation of results.

Response: Thank you for your valuable comments and suggestions. We have now highlighted the statistical significance of the observed differences in the main text, as well as in Figures 2, 7, 8, 9, and 10, with annotations marked in red text on a yellow background. The specific details are as follows:

Section 2.1: As depicted in Figure 2A and S2A, the DDP@18-crown-6 complex exhibited peak UV absorbance at 301 nm after 48 h of mixing and the highest concentration of DDP in the solution (p < 0.0001).

Figure 2. The optimization of preparation time (A) and rotation rate (B) of DDP@18-crown-6 complex (n = 3) and optical photograph of cisplatin dispersed or dissolved (C) in ultrapure water (a) or 18-crown-6 aqueous solution (b). ANOVA or t tests were utilized to determine whether there are significant differences between the data. P value style: *p < 0.05; **p < 0.01; ***p <0.001; ****p < 0.0001.

Section 2.7: The absorbance peaked at a molar fraction of 0.5 (p < 0.0001), indicating an optimal 1:1 binding ratio between DDP and 18-crown-6, consistent with molecular docking and phase solubility studies.

Figure7. Job’s plot (A) (n = 3) and binding structural schematic (B) of cisplatin and 18-crown-6 at Xm = 0.9. The C, N, O, H, Cl, and Pt atoms are colored in cyan, blue, red, white, green, and orange, respectively. Orange dashed lines are coordinate covalent bonds. Red dashed lines indicate electrostatic interaction and blue dashed lines are hydrogen bonds. ANOVA or t tests were utilized to determine whether there are significant differences between the data. P value style: *p < 0.05; **p < 0.01; ***p <0.001; ****p < 0.0001.

Section 2.8.1: As illustrated in Figure 8A, after 4 days, the concentration of DDP in the DDP@18-crown-6 complex solution remained at 99.08%, surpassing that in pure water (81.03%) (p < 0.0001) and 0.9% NaCl solutions (97.70%) (p = 0.0035). The DDP@18-crown-6 complex in 0.9% NaCl solutions exhibited the highest stability, with DDP concentration reaching 99.52% after 4 days (p = 0.0069), indicating optimal stability due to the combined effects of NaCl and 18-crown-6.

Section 2.8.2: To corroborate this conjecture, the impact of 18-crown-6 on the photostability of DDP was assessed. As depicted in Figure 9A, after 45 min of exposure to light, the degradation rate of DDP in the 18-crown-6 complex solution (2.49%) was substantially lower compared to that in pure water (7.55%) (p < 0.0001) and 0.9% NaCl solutions (4.31%) (p < 0.0001), with DDP content remaining at 97.51%. Furthermore, the presence of 0.9% NaCl further enhanced the photostability of DDP. Specifically, the DDP@18-crown-6 complex in 0.9% NaCl maintained 99.09% of DDP after 45 min of light exposure (p < 0.0001), indicating optimal photostability.

Figure 8. (A) Time-evolution stability of cisplatin (in H2O), cisplatin (in 0.9% NaCl aqueous solution), DDP@18-crown-6 complex (in H2O), and DDP@18-crown-6 complex (0.9% in NaCl aqueous solution), and (B) time-evolution stability of cisplatin at different 18-crown-6 concentrations (n = 3). ANOVA or t tests were utilized to determine whether there are significant differences between the data. P value style: *p < 0.05; **p < 0.01; ***p <0.001; ****p < 0.0001.

Figure 9. (A) Photostability of cisplatin (in H2O), cisplatin (in 0.9% NaCl aqueous solution), DDP@18-crown-6 complex (in H2O), and DDP@18-crown-6 complex (in 0.9% NaCl aqueous solution), and (B) photostability of cisplatin at different 18-crown-6 concentrations (n = 3). ANOVA or t tests were utilized to determine whether there are significant differences between the data. P value style: *p < 0.05; **p < 0.01; ***p <0.001; ****p < 0.0001.

Figure 10. In vitro cytotoxicity of cisplatin (A), DDP@18-crown-6 complex (A), and 18-crown-6 (B) (n = 3). ANOVA or t tests were utilized to determine whether there are significant differences between the data. P value style: *p < 0.05; **p < 0.01; ***p <0.001; ****p < 0.0001.

Q2. Figure 2 is challenging to assess from a substantive perspective due to the lack of information regarding the cisplatin solution concentration and the absence of a definition for the cisplatin complex with the complexing agent. There are three data series presented, but no clear relationships between the concentrations of the described systems can be discerned. How can the increase in the absorbance of the complex relative to pure cisplatin be interpreted if there is no clarity on the systems that were actually studied? Detailed information about the presented systems must be included in the figure description.

Response: Thank you for your valuable comments. We have revised the description in Figure 2 to include the relationship between cisplatin concentration and absorbance, along with the corresponding standard curve equation for cisplatin-absorbance. Additionally, we have more clearly defined the cisplatin complex with the complexing agent and explained the observed increase in absorbance for the complex relative to cisplatin. To better represent the changes in concentration within the system, we recalculated and converted the absorbance data in Figure 2 to cisplatin concentrations based on the standard curve, replacing absorbance values with concentration values. Furthermore, the original data on complex preparation time and optimal stirring speed based on absorbance have been moved to the Supplementary Material for reference. The additions have been highlighted in red with yellow in the body of the text. The details are as follows:

Section 2.1: DDP was configured as an aqueous solution at various concentrations. As illustrated in Figure S1, the concentration of the DDP solution exhibited a direct correlation with absorbance at 301 nm. The resulting linear regression analysis yielded the equation Y = 0.3459X + 0.0827 (R² > 0.99). Upon the addition of 18-crown-6 to the DDP solution, a potential increase in solubility was observed, likely due to the formation of a complex between DDP and 18-crown-6. Absorbance measurements of the resulting complex system were then performed, and the standard curve equation was applied to assess the impact of the DDP@18-crown-6 complex on the solubility enhancement of DDP.

The duration of complex preparation markedly influenced the binding affinity between DDP and 18-crown-6. Optimal preparation time was thus evaluated [25]. As depicted in Figure 2A and S2A, the DDP@18-crown-6 complex exhibited peak UV absorbance at 301 nm after 48 h of mixing and the highest concentration of DDP in the solution (p < 0.0001).

Figure 2. The optimization of preparation time (A) and rotation rate (B) of DDP@18-crown-6 complex (n = 3) and optical photograph of cisplatin dispersed or dissolved (C) in ultrapure water (a) or 18-crown-6 aqueous solution (b). ANOVA or t tests were utilized to determine whether there are significant differences between the data. P value style: *p < 0.05; **p < 0.01; ***p <0.001; ****p < 0.0001.

Figure S2. The optimization of preparation time (A) and rotation rate (B) of DDP@18-crown-6 complex by UV absorbance at 301 nm (n = 3). ANOVA or t tests were utilized to determine whether there are significant differences between the data. P value style: *p < 0.05; **p < 0.01; ***p <0.001; ****p < 0.0001.
